# Engineering flexible superblack materials

Yucheng Yang[1], Botond Sánta[1], Ashok Ponnuchamy[2], Edward C. Kinzel[1], Anthony J. Hoffman[2] & Matthew R. Rosenberger ®[1] ✉

Flexible superblack materials are crucial for minimizing stray light, complicating object identification, and serving as low reflectance standards. However, the applications of existing superblack materials are limited due to challenges related to cost-effective scalable manufacturing, surface durability, and material conformability. Furthermore, existing fabrication platforms struggle to tailor superblack materials to application-specific needs. This work introduces an engineering platform that combines silicon mold fabrication and polymer casting to produce flexible superblack materials. This platform achieves repeatable wafer-scale production of superblack materials with a minimum reflectance of 0.15% and less than 0.4% across the visible spectrum. The sample reflectance is weakly dependent on illumination angles from 0° to 50° and observer angles from 0° to 70° when the illumination angle is less than 20°. This Lambertian-like reflectance profile enables the material to effectively conceal three-dimensional features in digital images even under intense lighting conditions. This platform can engineer the material surface to withstand tweezer scratches without significantly compromising its reflectance properties. This work introduces an engineering platform for designing flexible superblack materials, addressing key challenges in scalability, surface durability, mechanical flexibility, and customization.

The development of low reflectance materials is essential for a broad array of applications in the visible portion of the electromagnetic spectrum ($\lambda \sim 380$–$700\,\text{nm}$). Such materials play central roles in absorbing stray light[1,2], preventing or complicating object identification using visible light[3], and serving as low reflectance standards[1]. Key to these applications is demonstrating materials that exhibit low total reflectance overall all directions, $R_{Tot}$, (referred to as the hemispherical reflectance) for all angles of illumination across the visible spectral range. Additionally, the light should be reflected isotropically, known as Lambertian reflection, to mitigate specular reflections that can lead to glare. The combination of broadband, low $R_{Tot}$, and Lambertian reflection results in omni-directional, superblack[4] ($R_{Tot} < 0.4\%$) materials that are effective regardless of the illumination and observation angles. In addition to being superblack, many applications also require other material attributes such as surface durability, reusability, conformability, and manufacturability. Consequently, developing a versatile platform for engineering application-specific material properties

(e.g., optical and mechanical) is critical for enabling and advancing a wide range of applications.

Although superblack materials have been demonstrated using a variety of approaches, their usefulness is limited due to challenges related to material robustness[5–8], cost-effective manufacturing scalability[1,2,4], and material conformability[5–7]. Furthermore, the ability to tailor the properties of superblack materials for specific applications is significantly limited in most previous demonstrations. These limitations confine the use of existing superblack materials mainly to calibration references or for use in laboratory and artistic demonstrations. For instance, commercially available materials like Spectralon exhibit near-perfect Lambertian low reflectance ($R_{Tot} \sim 1\%$), but their high costs limit their use in large-scale applications. Furthermore, the reflectance of the blackest commercially available version of Spectralon is still relatively high compared to other alternatives, including the materials presented here. While vertically aligned carbon nanotubes[5,7,9] (VACNT) and black silicon[6,10,11] exhibit low reflectance

[1]Department of Aerospace and Mechanical Engineering, University of Notre Dame, Notre Dame, IN, USA. [2]Department of Electrical Engineering, University of Notre Dame, Notre Dame, IN, USA. ✉e-mail: mrosenb2@nd.edu

($R_{Tot} < 0.1\%$ to ~0.2%) in the visible range with a near-Lambertian reflectance distribution, their nanoscale vertical structure makes them extremely fragile. Physical contact drastically changes the reflectance of these materials[1,2].

Recently, supreme black materials were demonstrated using high-energy heavy ion track-etched plates as templates to produce touch-proof flexible materials with $R_{Tot} < 0.1\%$[1]. However, these materials exhibit strong retroreflection (light is reflected backwards along the illumination direction) for certain angles of illumination[1]. Additionally, the production of the mold requires expensive equipment (high-energy heavy ion accelerators[1,2]), limiting scalability and accessibility. While mold replication presents a pathway towards scalability, there are open questions related to mold longevity and production consistency. The pathway to achieving high-volume manufacturing using track-etched molds remains uncertain.

In this work, we present an approach that leverages conventional silicon fabrication tools and polymer casting techniques to produce flexible, broadband, omni-angle, superblack materials. We demonstrate repeatable silicon mold fabrication on 4-inch wafers and uniform polymer casting, establishing a feasible pathway for large-scale manufacturing. We fabricate and characterize a series of microcavities in polydimethylsiloxane (PDMS) using the silicon molds, demonstrating $R_{Tot} < 0.4\%$ across the visible portion of the spectrum and a minimum $R_{Tot} = 0.15\%$ at 450 nm for near-normal illumination. For 450 nm illumination at higher angles of incidence, $R_{Tot}$ ~ 0.5% at $\theta_i = 60°$ and ~2% at 75°. We further demonstrate the importance of a platform for engineering the PDMS mold by designing superblack materials with enhanced mechanical robustness that are capable of withstanding scratching by tweezers with up to 900 kPa of pressure without significantly compromising their superblack reflectance. Finally, to show the utility of a centimeter-scale, flexible, omni-angle, superblack material, we conceal the geometry of a three-dimensional (3D) object by covering it partially with the material. This platform provides opportunities for designing and manufacturing flexible superblack materials where the optical and mechanical characteristics and robustness of the materials can be engineered.

## Results

### Reflectance and microcavity design

Superblack material reflectance is complex, and the traditional Phong model does not accurately portray the sample reflectance. The Phong model[12], which has been used in existing literature[1], describes reflected light in terms of specular reflection, diffuse reflection, and ambient reflection components. The angles of the incident light direction ($\hat{\mathbf{i}}$) and reflected light direction ($\hat{\mathbf{r}}$), $\theta_i$ and $\theta_r$, respectively, are specified relative to a global, macroscale surface normal, $\hat{\mathbf{n}}$, Fig. 1a. For simple glossy materials, specular reflection where $\theta_i \approx \theta_r$ is the dominant component. The Phong model fails to accurately describe many real materials, which often exhibit glancing angle reflection and retro-reflection. Glancing angle reflection occurs on rough surfaces[13,14] (or microfacets) where light is preferentially reflected at glancing angles and strongly depends on $\theta_i$. Retroreflection[1] has been observed in microcavity samples under glancing $\theta_i$. For engineered superblack materials, the dominant reflectance component is likely at a location where $\theta_i \neq \theta_r$. Instead, light is reflected into off-normal directions, resulting in a black appearance when observed from near-normal incidence. Therefore, both careful engineering and rigorous characterization of the reflection are essential for developing and validating omni-angle superblack materials.

To engineer omni-angle superblack materials, we develop a simple phenomenological model for $R_{Tot}$ that relates geometry of the designed materials to the various components of reflected light (Fig. 1b). The key geometric parameters include the slope of the cavity entrance ($S_E$), slope of the cavity body ($S_B$), cavity sidewall smoothness, bottom surface area, and top surface area. The general design strategy is to leverage many lossy reflections inside of the PDMS microcavities to minimize $R_{Tot}$. This strategy results in several design rules. The combination of smooth sidewalls and large $S_B$ results in light traveling deep into the microcavity, which leads to more lossy reflections in the microcavity and lower $R_{Tot}$[1,2]. Controlling the sidewall roughness is important as surface roughness would result in diffusive reflection out of the microcavity[1,2]. Reducing the top surface area lowers reflection for all angles of incidence[4] (red arrows). Similarly, reducing the bottom surface area lowers glossy reflection from this region of the microcavity that can escape after a single reflection when the incident light is a near-normal angles[4] (blue arrows). Finally, increasing $S_E$ reduces retroreflection for glancing angle illumination (green arrows). In order to repeatably manufacture such a microcavity, a casting mold with the inverse geometry is needed[15], as shown in Fig. 1c.

### Microcavity mold and microcavity fabrication

We implement these design rules by fabricating high aspect ratio hexagonally-close-packed cone molds using conventional silicon fabrication tools[16–22]. The fabrication process, depicted in Fig. 2a, uses photolithography to define a mask for deep reactive ion etching[17,23–25] (DRIE). Subsequently, we implement a tapered DRIE process to create sharp trenches, minimizing the top surface of the microcavity[23]. Following this, an oxygen plasma ashing removes residual photoresist and polymer that is introduced during the DRIE passivation step. Isotropic etching is performed at a higher vacuum pressure than that used in the

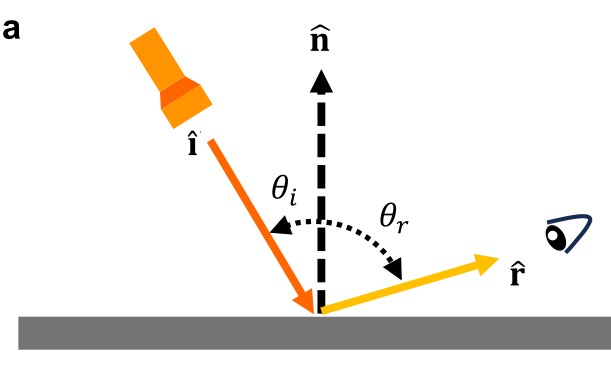

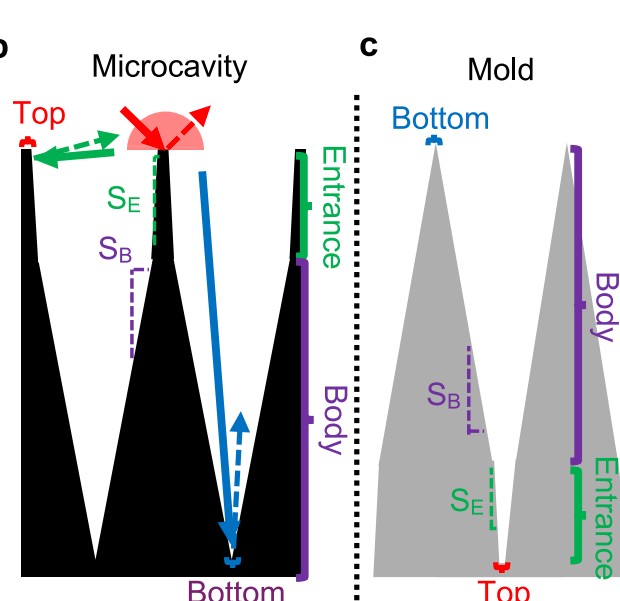

**Fig. 1 | Reflectance dependence diagram and microcavity/microcavity mold design for light trapping applications. a** reflectance dependence diagram. $\theta_i$ is the angle between $\hat{\mathbf{n}}$ and incidence of illumination ($\hat{\mathbf{i}}$). $\theta_r$ is the angle between $\hat{\mathbf{n}}$ and incidence of reflection ($\hat{\mathbf{r}}$). **b** cross-section of a microcavity and **c** cross-section of the mold that will produce the microcavity.

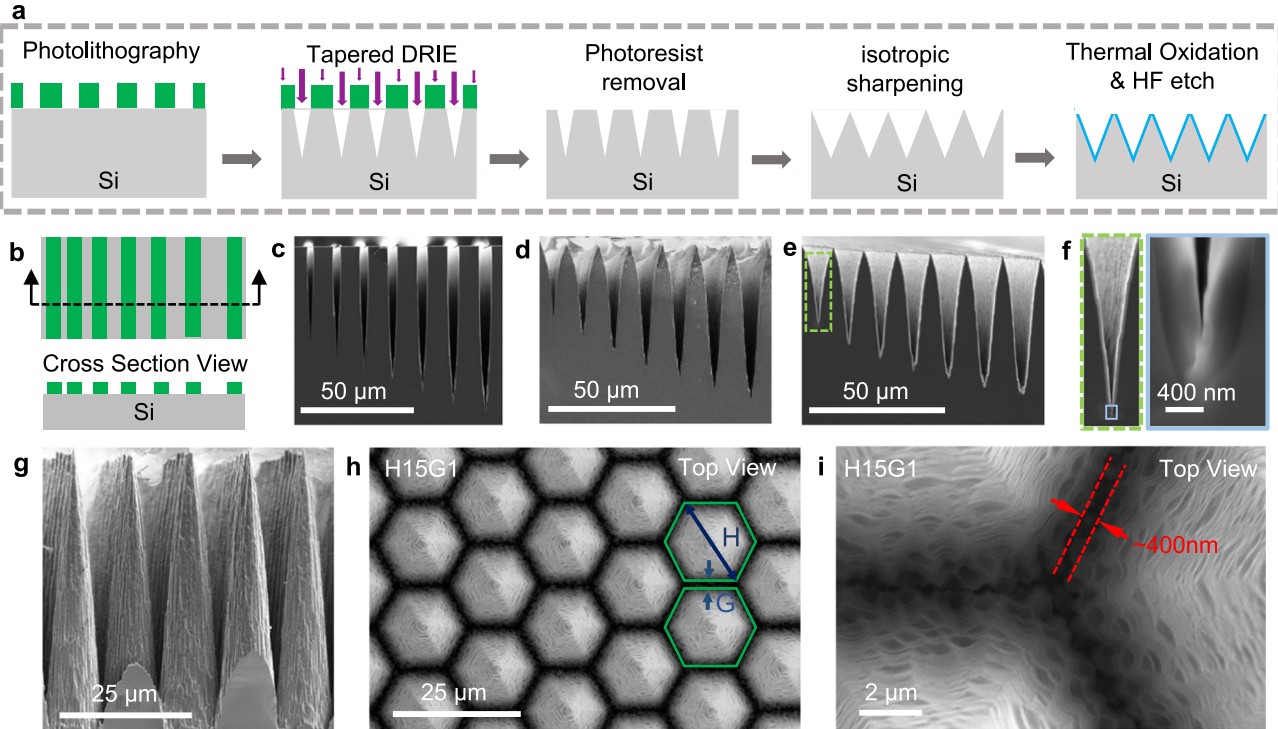

**Fig. 2 | The microcavity mold fabrication process. a** The silicon fabrication process flow for the microcavity mold. **b** photolithography line test patterns with varying gap for developing the fabrication process. SEM cross-sectional image of the line patterns after (**c**) tapered DRIE (**d**) isotropic sharpening (**e**) oxidation. The gap changes from left to right from 1 μm, 1.5 μm, 2 μm, 2.5 μm, 3 μm, 3.5 μm, 4 μm. **f** a zoom-in SEM cross-sectional image of the 1 μm gap. **g** SEM cross-sectional image of densely packed silicon pillars. Image of H15G1 (H15G1: Hexagon diameter (H) = 15 μm and Gap distance (G) = 1 μm) **h** top view **i** zoom-in silicon bottom.

DRIE process to form sharp edges at the top of the mold while minimizing material removal at the bottom[16]. Wet oxidation is used to further reduce the bottom surface area of the mold. Finally, a buffered oxide etching step smoothens the silicon mold surface[26], allowing for realization of samples with smooth sidewalls and assisting with release of the PDMS from the silicon mold.

We demonstrate the fabrication steps of the silicon mold using a test structure with 10 μm wide lines of increasing spacing from 1 μm to 4 μm, Fig. 2b. Figure 2c–f shows cross-sectional images obtained using scanning electron microscopy (SEM) of the test structure. Figure 2c shows the line patterns after the tapered DRIE and photoresist removal. Sharp trenches with an angle of taper of ~5° at the bottom of the trench for a 1 μm opening are observed. The bottom of these trenches will become the top surface of the microcavity with high $S_E$. Figure 2d shows the result of the sharpening step that produces a thin peak at the top surface while maintaining the small bottom surface of the trench realized in the previous fabrication steps. Figure 2e shows the conformal oxide formed on the mold structures. Figure 2f shows a zoom-in view of the 1 μm trench after growth of 400 nm oxide which resulted in a sharp bottom.

By changing the photolithography patterns from lines to hexagons, we fabricate dense, high aspect ratio silicon pillars (Fig. 2g). The hexagonal photolithography pattern design consists of two parameters: hexagon diameter (*H*) and gap distance (*G*) (see inset cartoon in Fig. 2h and Supplementary Fig. 1a). Figure 2h shows a top-view SEM image of H15G1 (*H* = 15 μm and *G* = 1 μm). The bottom surface width of H15G1 pattern is measured to be ~400 nm (Fig. 2i). By changing G from 1 μm to 4 μm, we can alter the microcavity mold geometry, as seen in Supplementary Fig. 1b and Supplementary Fig. 1c.

We produce flexible microcavity samples by drop casting PDMS (1:8 reagent to base ratio[27]) mixed with 1% nigrosine onto the silicon molds as shown in Fig. 3a. The molds are coated with trimethyl-chlorosilane (TMCS) prior to drop casting to improve PDMS release and increase silicon mold longevity[28]. After casting, the PDMS is cured on the microcavity mold at 100 °C for 20 min. The PDMS is then released from the microcavity mold. Based on SEM inspection of a region greater than 4 mm[2], no significant PDMS residues were observed after more than five casts (Supplementary Fig. 2 and Supplementary Fig. 3). This is essential for reuse of the mold and represents progress towards a scalable fabrication process. Finally, the PDMS surface is treated with UV-ozone to reduce adherence of ambient dust onto the sample surface[29]. Importantly, PDMS is an elastomer that differs from materials like carbon nanotubes[5,7,9], silicon[6,30], and carbon coatings[8] by offering enhanced surface robustness and bulk flexibility.

We demonstrate mold uniformity and repeatability by fabricating three duplicate 4-inch wafer-scale silicon molds for two different lithography pattern designs, H15G1 and H10G1 (Fig. 3b). The patterns are highly uniform across the entire 4-inch wafer and consistent between wafers. We show the reusability of the mold by casting a H10G1 sample three consecutive times on the same mold. Under intense illumination of ~19,000 lx, and 1/50 s shutter speed, there is no strong reflectance variation between consecutive castings (Fig. 3c). Under this illumination and shutter speed, a reflectance variation of ~0.1% or more is observable (Supplementary Fig. 4) suggesting no clear degradation of the silicon molds after three consecutive castings. Additionally, we performed quantitative measurements of reflectance for five castings from the same mold region and found a total range of values of 0.03% reflectance (approximately ±10% variation or less from the nominal value) for total reflectance, quantifying the repeatability of our casting process (Supplementary Fig. 5).

The casted PDMS using mold H15G1 in Fig. 3d shows characteristics of an efficient optical trapping microcavity based on the microcavity design discussed in Fig. 1b. At 30° tilted view, the slope of the entrance cavity is seen to be near vertical, and the sidewall roughness is nanoscale smooth (Fig. 3e). Figure 3f shows sub-visible wavelength top

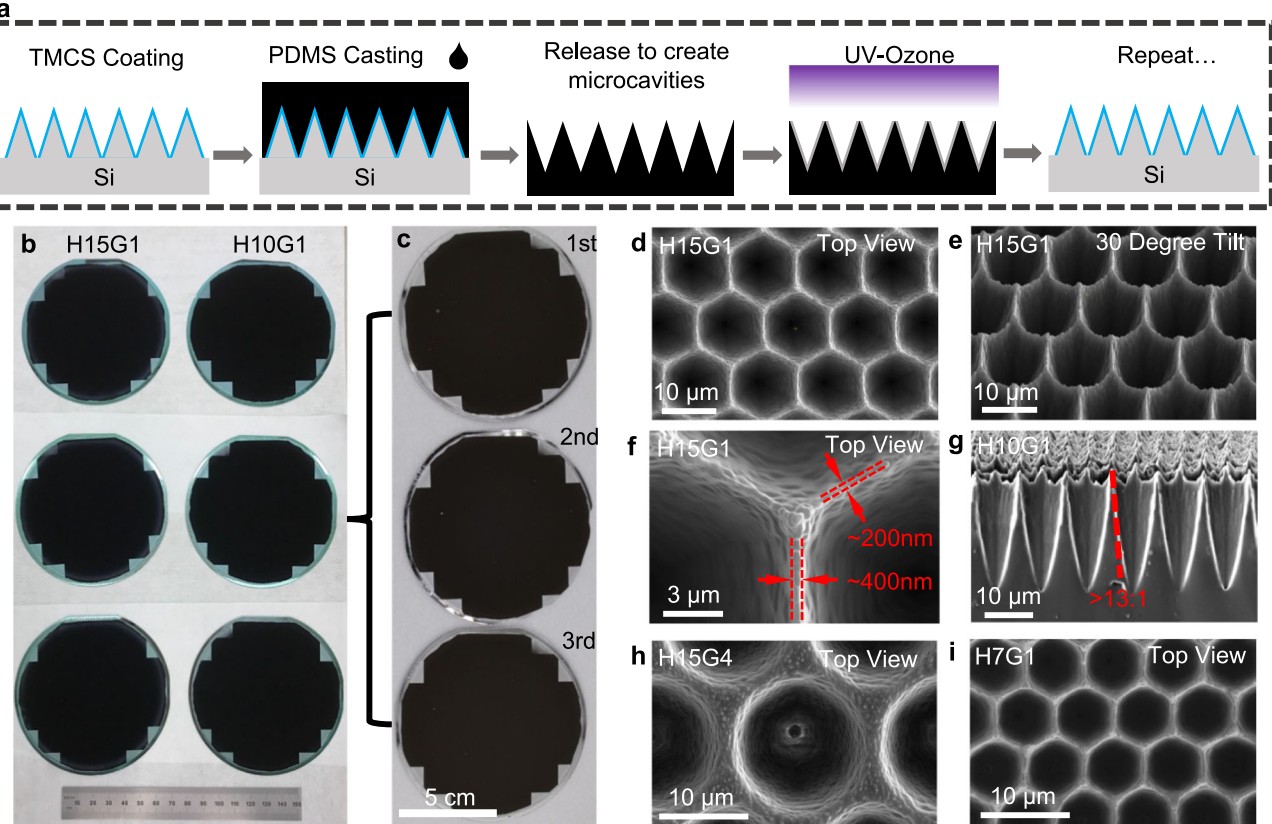

**Fig. 3 | Repeatable PDMS microcavity fabrication process. a** The PDMS micro-cavity casting process flow. **b** demonstrating silicon mold fabrication repeatability and uniformity for both H15G1 and H10G1. The wafers are on top of cellulous composite cleanroom wipes. The image is taken inside of a solvent hood. Blue color around the silicon wafer is due to the grown silicon dioxide. **c** demonstrating repeatable PDMS casting and silicon mold reusability/longevity. All three casts are

from one H10G1 mold as shown. The image is taken in a light box under 19,000 lx illumination with a shutter speed of 1/50 s. SEM images of PDMS microcavity using silicon mold H15G1: **d** top view, **e** 30° tilt, **f** zoom-in top view. **g** SEM cross-sectional view of H10G1. SEM top view of PDMS microcavity using silicon mold with **h** different gap distance, H15G4, and **i** different hexagon diameter, H7G1.

surface width, varying from ~200 nm at thin region to ~400 nm at the thick region. From a cross-sectional view of casted PDMS from mold H10G1 (Fig. 3g), $S_E$ is measured to be ~4°, resulting in an aspect ratio of >13, which exceeds the reported value of ≥ 4[1] or > 5[2] in the existing literature. Steep $S_E$ should result in better glancing angle performance. Furthermore, the zoomed-out SEM image demonstrates that the microcavity is uniform across mm-scale areas (Supplementary Fig. 6).

Furthermore, the microcavity features can be engineered by changing the photolithography patterns and etching parameters to modify the optical and mechanical properties of the material, distinguishing it from track etching methods[1]. By changing hexagon pattern parameters ($H$ and $G$), we can engineer the microcavity mold geometry (Supplementary Fig. 1c). By increasing the gap distance, we observed an increase in the microcavity side wall thickness (Fig. 3h). We can change the microcavity aspect ratio and hexagon opening by reducing the hexagon diameter (Fig. 3i). Later, we further demonstrate that the top surface geometry of the microcavity can be modified for enhanced mechanical robustness by modifying the etching process.

**Reflectance characterization**

We employ three quantitative techniques to characterize the reflectance from the fabricated samples: side port $R_{Tot}$ integrating sphere measurement, center mount angle-dependent $R_{Tot}$ integrating sphere measurement, and two-dimensional-bidirectional reflectance distribution function (2D-BRDF) measurement. The side port $R_{Tot}$ integrating sphere measurement provides $R_{Tot}$ of a sample at $\theta_i = 5°$ (Supplementary Fig. 7a). The center-mount integrating sphere provides the $R_{Tot}$ of a sample as a function of $\theta_i$ (Supplementary Fig. 7b).

The 2D-BRDF measurements characterize the angle-resolved reflective radiative intensity in the plane of incidence as a function of $\theta_r$ and $\theta_i$ by using two rotation stages with the same axis of rotation to control the orientation of the sample and detector with a fixed source position (Supplementary Fig. 7c). All of these techniques involve highly directional illumination setups, distinct from ambient lighting conditions. Light box digital photographs enable comparative reflectance analysis of samples under non-directional illumination, mimicking what human eyes see under ambient illumination (Supplementary Fig. 7d).

We use spectrally resolved side port $R_{Tot}$ integrating sphere measurement to study the role of the gaps between hexagons and the size of the hexagons for fixed processing conditions on the total reflection. Figure 4a shows that for decreasing $G$ (from 4 μm to 1 μm) and a constant hexagon size ($H = 15$ μm), a monotonic reduction of total reflectance is observed. The lowest reflectance microcavity sample, H15G1, exhibits a total reflectance as low as 0.15% at a wavelength of around 450 nm which is lower than fabricated black silicon (Black Si), a known effective optical absorber[6]. The SEM image of black Si is shown in Supplementary Fig. 8. The reduction in the reflectance with decreasing $G$ is likely due to the reduction of the top surface area as shown in SEM images (Supplementary Fig. 9a). We further confirmed the change in top surface area with optical surface topography measurements using coherent scanning interferometry (Supplementary Fig. 9b).

To understand the impact of microcavity high-to-width ratio ($h/w$) on the microcavity reflectance, we vary $H$ for constant $G$ under fixed processing conditions. Here, $G$ is 1 μm and $H$ is varied from 7 μm ($h/w = ~7$) to 15 μm ($h/w = ~4$), (Supplementary Fig. 10 and Supplementary

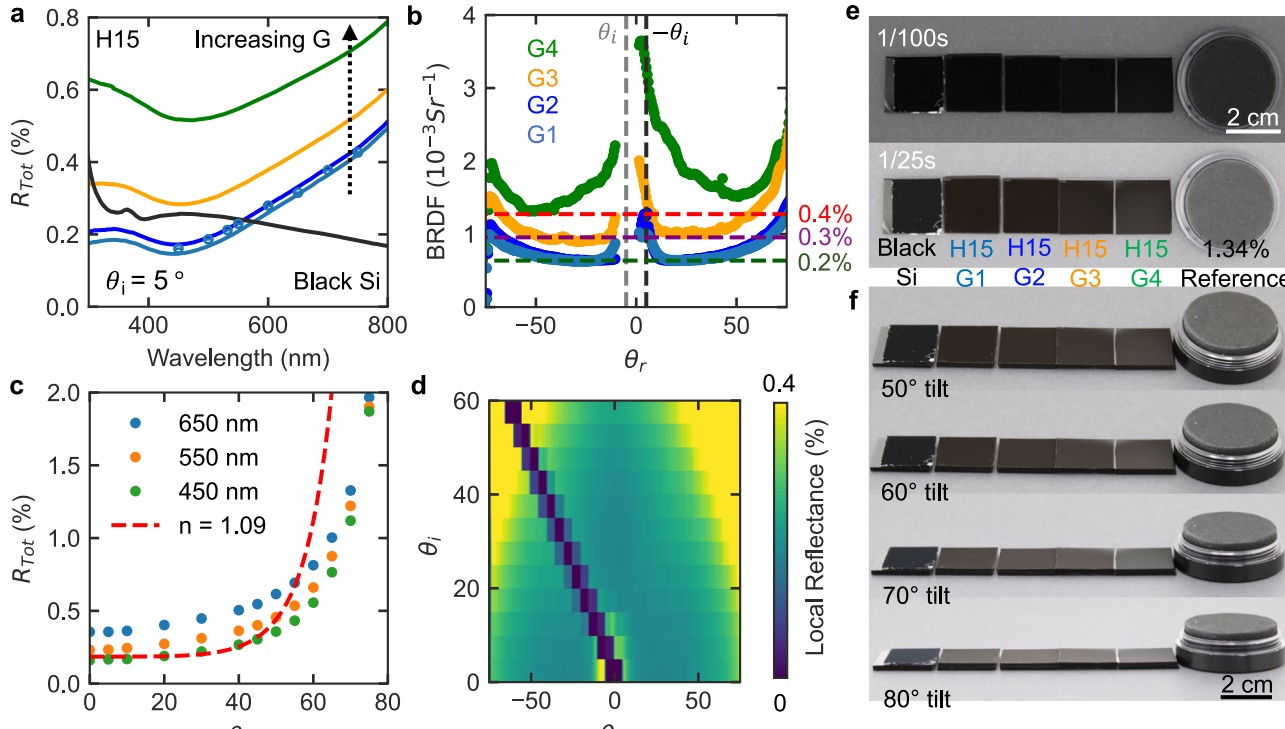

**Fig. 4 | Optical characterization. a** Spectrally resolved integrating sphere measurement of microcavity samples with fixed $H$ of 15 μm and varying $G$ (1 μm, 2 μm, 3 μm and 4 μm). **b** 2D-BRDF measurement of PDMS microcavity samples with constant $H$ and varying $G$. The vertical gray and black dotted line are the $\theta_i$ of the laser and specular angle, respectively. The color-coded horizontal dashed lines are the hypothetical perfect Lambertian reflectance sample with reflectance percentage as labeled. The region without data is due to the detector blockage of the laser. **c** Angle dependent integrating sphere measurements from $\theta_i = 0°$ to $\theta_i = 75°$ compared with a material with an effective $n$ of 1.09 (red-dashed line).

**d** 2D-BRDF characterization with $\theta_i$ from 0° to 60° and $\theta_r$ from −75° to 75°. The region with 0% location reflectance is due to the detector blockage of the laser. **e** Top view of black silicon, PDMS microcavity samples with varying gap size, and diffuse reference standard inside a lightbox under ~20,000 lx illumination using camera shutter speeds of 1/100 s and 1/25 s. **f** Off-normal view under ~19,000 lx illumination using 1/25 s shutter speed. The change of illumination comparing to the top view is due to that the front white panel is removed. The background of the photos is black photo panel.

Table 1). Only small changes in the total reflectance are observed (Supplementary Fig. 11a). Based on the side port integrating sphere measurements, the top surface area is more important for reducing reflectance than the cavity aspect ratio for the investigated designs when the aspect ratio is greater than ~4.

Both the accuracy and precision of the reflectance characterization for an ultralow reflectance material is critical. We performed two characterizations to validate the accuracy of integrating sphere measurements. First, we compared our reference standard measurement results with those characterized by a third-party detailed in the "methods" section (Supplementary Fig. 12a and Supplementary Fig. 12b). Our measurements are nominally consistent with the third-party results. Second, we performed the reflectance characterization on H15G1 using a second side port integrating sphere measurement setup with a 150 mm integrating sphere, a tunable laser source, and independent detector. The measurement results are depicted in Fig. 4a as dots. The reflectance measurements of H15G1 shows nominal consistency across the two independent integrating sphere setups with different light source, detector, and integrating sphere. The precision of both integrating sphere measurement is ~100 times lower than H15G1 reflectance as shown in Supplementary Fig. 12c and Supplementary Table 2. These experiments validate the qualitative and quantitative reflectance values in our work.

To understand the angle-dependent reflectance from the samples, we use a goniometer to measure the BRDF (units of Sr⁻¹) of the microcavity samples as a function of $\theta_r$ from 75° to −75° when the sample is under $\theta_i = 5°$ illumination (measurement accuracy is shown in Supplementary Fig. 13). We perform BRDF measurements on

samples with fixed $H$ of 15 μm and varying $G$ (Fig. 4b). We observe an increase in the BRDF at every $\theta_r$ as $G$ increases. More importantly, when the gap distance is less than 2 μm, there is no prominent broadened specular reflection compared to H15G4, which shows a clear increase near the specular reflection angle. The small increase of specular reflectance shown in H15G1 likely is due to the bottom surface, which is discussed further later. When we measured samples with fixed $G$ of 1 μm and varying $H$ (7 μm, 10 μm, and 15 μm), we observe only small changes in the BRDF (Supplementary Fig. 11b). Under 532 nm laser illumination at $\theta_i = 5°$, the lowest reflectance microcavity sample, H15G1, exhibiting reflectance <0.4% for $\theta_r$ ranging from −70° to 70° (red dashed line in Fig. 4b).

Angle-dependent integrating sphere measurements measure the total reflectance as a function of $\theta_i$. We illuminate the sample using a tunable light source at 650 nm, 550 nm, and 450 nm (measurement noise shown in Supplementary Table 3). With increasing $\theta_i$, we observe an increase in total reflectance (Fig. 4c). However, the rate of increase is significantly lower compared to existing literature[1]. At $\theta_i = 75°$, we observe increases of only ~5, 8, and 11 times compared to the total reflectance at $\theta_i = 5°$ for illumination wavelengths of 650 nm, 550 nm, and 450 nm, respectively (Supplementary Fig. 14a). This represents a significant improvement compared to the ~100× increase in the total reflection reported in literature[1]. We attribute this improvement to the higher aspect ratio at the microcavity entrance achieved in our material. The aspect ratio of our superblack microcavity at the entrance is > 13 ($S_E = 4.3°$) (Fig. 3g), in contrast to the previously reported value of ≥ 4 ($S_E = 14.0°$)[1] or > 5 ($S_E = 11.3°$)[2]. This improvement is limited to relative changes in the reflectance as

a function of $\theta_i$ rather than the absolute reflectance. The red dashed line represents a hypothetical material with an effective refractive index ($n$) of 1.09 that is calculated with unpolarized Fresnel's equations. The $n$ is chosen such that the curve matches the flexible superblack microcavity reflectance at near-normal incidence $\theta_i \sim 0°$. (Supplementary Fig. 14b, c). Here, we demonstrate that the near glazing angle reflectance of our superblack microcavity significantly outperforms a material[31] with $n = 1.09$. Thus, the dependence of total reflectance on $\theta_i$ for our omni-angle superblack microcavity is weaker than that of both supreme black materials[1] and bulk materials with similar reflectance at near-normal incidence[30,31].

We use 2D-BRDF measurements (Fig. 4d) to quantify the directional components of the reflected light for various incident angles. Here, we vary $\theta_i$ by rotating the sample from 0° to 60° with a step size of 5°. We change the position of the detector to control $\theta_r$ from −75° to 75° with a step size of 0.5°. To present the data in terms of reflectivity, the corresponding local reflectance is calculated by multiplying BRDF by $\pi$. The resultant values represent the total outgoing light flux from a surface, assuming it is perfectly Lambertian (i.e., reflects light equally in all directions). The raw BRDF and BRDF * Cos($\theta_r$) are shown in Supplementary Fig. 15. When $\theta_i < 25°$, the sample remains superblack within all measured $\theta_r$. Although increasing $\theta_i$ beyond 25° increases the reflectance at grazing detector angles, the sample remained superblack, varying from ~0.2% to 0.4%, within most of $\theta_i$ and $\theta_r$. Besides the glossy components at small $\theta_i$ from the bottom of the microcavity, we observe no specular or broadened retroreflective components from the entrance of the microcavity at high $\theta_i$. This glossy components at small $\theta_i$ are also present in some other superblack materials, such as black silicon[6] and VACNT[7].

We compare our fabricated microcavity samples with black silicon and a Lambertian reflectance standard ($R_{Tot} = -1.4\%$, Supplementary Fig. 12b) using digital images acquired with various shutter speeds in a lightbox with a black photo panel underneath (Fig. 4e). To highlight differences between the samples, we change the shutter speed to simulate changes to the lighting conditions. For example, decreasing the shutter speed, increases the exposure time, which simulates stronger illumination. The black photo panel as a function of luminous flux per area (lx) is shown in Supplementary Fig. 16. Figure 4e shows the H15 series of samples. Here, the perceived blackness of the samples is consistent with the integrating sphere measurements; increasing gap size results in higher reflectance. Additionally, the relative spectral reflectance can be observed in the photographs as sample H15G1 exhibit a slight red hue due to its higher reflection at longer wavelengths. Black silicon appears black, consistent with its roughly uniform reflectance across all visible wavelengths.

Digital images are also used to examine the sample reflectance under non-directional illumination for different observation angles, $\theta_r$. All these images are captured under 19,000 lx illumination with a shutter speed of 1/25 s to observe small reflectance changes (Fig. 4f). There are no significant changes in the sample appearance between 50° and 60° camera tilt. However, for larger angles, 70° and 80°, the sample appears brighter for both microcavities and black silicon. In contrast, the -1.4% Lambertian standard appears nominally the same for all observation angles. Under non-directional illumination, we observe an even weaker dependence between sample reflectance and $\theta_r$ comparing with highly direction illumination Fig. 4c. In this demonstration, within 60° of camera tilt, there is no significant change in the visible reflectance. This result suggests that the 1 μm gap flexible superblack microcavity exhibits near-Lambertian characteristics for $\theta_r < 60°$ under non-directional illumination.

## Superblack material applications
We test the microcavity robustness by performing touching, water immersion, water sonication, and air gun tests. We apply a normal load >600 kPa (>180 times greater than previous work[1]) onto the sample

and observe no clear changes to the sample reflectance at the region where the stress was applied (Supplementary Fig. 17a, b). We measure the reflection before and after a water immersion[4] test followed by air gun drying test[1,4]. We do not observe significant changes to the sample reflectance at the region where the water immersion and air blowing was performed (Supplementary Fig. 17c). Additionally, we perform water sonication by placing the sample in an ultrasonic cleaner for 3 min and observed a ~0.03% increase in reflectance at 450 nm wavelength (Supplementary Fig. 17d). To investigate the failure modes of the microcavities, we drag a tweezer across the sample surface, similar to previous studies[1]. We observe an increase in the sample reflectance after damaging the sample. We use SEM imaging to identify two mechanisms that cause the increase in reflection: (1) removal of the sharp top surface and (2) full or partial blockage of intact microcavities (Supplementary Fig. 18). To accommodate applications where the robustness is a primary concern, the fabrication process can be modified to improve the mechanical robustness. Unlike existing strategies for creating superblack surfaces, our platform allows us to easily engineer our molds and the resultant microcavities to achieve targeted optical and mechanical properties for specific applications.

We demonstrate the design flexibility of our approach by engineering superblack materials for applications requiring mechanical robustness. We accomplish this by modifying the tapered DRIE process, adjusting the passivation-to-etching step time ratio from 2:10 to 2:7. This process modification slightly reduces $S_E$ while maintaining a small top surface width, measured to be 450 nm (Fig. 5a). We evaluate the durability of the robust-H15G1 microcavities, named R-H15G1, using the metallic tweezer scratch tests and #2000 grit sandpaper. After damage testing, the sample reflectance is characterized using digital imaging under 20,000 lx illumination and varying shutter speeds (Supplementary Fig. 19). At a shutter speed of 1/100 s, no visible damage is observed from either the tweezer or sandpaper tests (Fig. 5b). However, at 1/50 s shutter speed, which mimics stronger illumination, scratches from the sandpaper test under a 300 kPa load are noticeable. Interestingly, scratches from the tweezer test under a 900 kPa load were not discernible until the shutter speed was increased beyond 1/10 s (Fig. 5c). Scratching with the tweezers predominantly results in collapse of the microcavities due to high shear loads, affecting a limited number of microcavities and resulting in minimal increase in the reflectance (Fig. 5d, e). Further analysis shows two different dominant mechanisms that contribute to the increased reflectance in the sandpaper scratching test: (1) abrasive material becoming embedded within the microcavity (Supplementary Fig. 20), and (2) abrasive material cutting through the microcavity (Fig. 5f, g).

We conducted additional surface robustness tests based on methods present in the existing literature: finger touching[1,4], dust roller[1], large-area tweezer scratching[2], and hydrophobicity measurement, to further highlight the advantages of our superblack microcavity surfaces. For the finger touching test, we apply force to the R-H15G1 surface by pressing with a finger six times, with a maximum applied load exceeding 360 g (the maximum measurable by the weighing scale), resulting in an estimated pressure of ~18 kPa, more than three times higher than previously reported ~3 kPa[1] and ~5 kPa[4] (Fig. 5h, Supplementary Movie 1). In the dust roller test, the superblack surface is subjected to six passes with a Scotch adhesive dust roller, also with a maximum applied load exceeding 360 g, generating an approximate pressure of 8 kPa. We note that the dust roller used here contains Scotch adhesive, rather than a silicone dust roller which was used in the existing literature[1] (Supplementary Movie 2). We compare tweezer scratching directly between black silicon and R-H15G1. After applying only a 20 g load, resulting in a pressure of ~65 kPa, the black silicon surface shows clear damage. In contrast, R-H15G1 can withstand a load of nearly 150 g (Fig. 5j), corresponding to a pressure of about 500 kPa, without significant degradation of reflectance properties (Fig. 5k). The video for the tweezer scratching test is included in

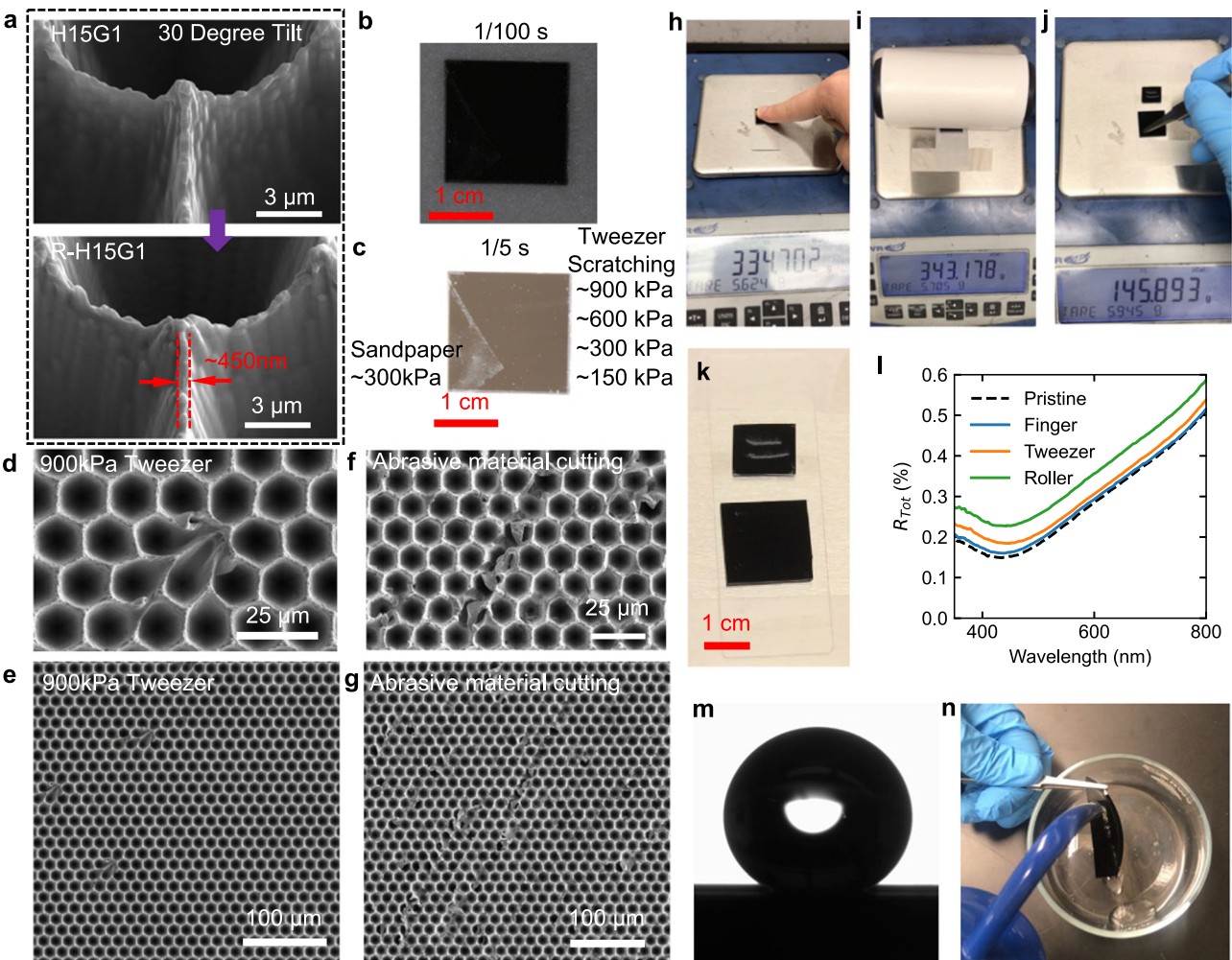

**Fig. 5 | Demonstration of robustness of R-H15G1. a** Microcavity $S_E$ is changed due to modifying etching parameters. The damaged R-H15G1 (sandpaper and metallic tweezer scratching) is shown under ~20,000 lx illumination with a shutter speed of **b** 1/100 s **c** 1/5 s. **d, e** are SEM images of the 900 kPa tweezer-scratched region. **f, g** are SEM images of abrasive material cutting due to ~300 kPa sandpaper scratching. Surface robustness was tested using **h** finger touching, **i** a Scotch adhesive dust roller, and **j** tweezer scratching. **k** A zoomed-in image shows the damage caused by tweezer scratching on black silicon, with no clear damage on R-H15G1. **l** $R_{Tot}$ measurements after the surface robustness test. **m, n** demonstrate the hydrophobic nature of the superblack microcavity surface.

Supplementary Movie 3. Figure 5l presents the $R_{Tot}$ measurements following each surface robustness test. Both finger and tweezer test do not significantly change R-H15G1 reflectance (<0.05%). We observe a ~0.08% increase in the reflectance after the Scotch adhesive dust roller test. We suspect the damage may be from the Scotch adhesive as the applied pressure is the lowest for the Scotch adhesive dust roller test.

Additionally, we performed contact angle measurements on the microcavity surface. Six measurements were taken at different surface regions, yielding an average contact angle of 138° with a standard deviation of 4°. The maximum contact angle recorded was 141° for a 3 µL water droplet (Fig. 5m). To further validate the hydrophobic properties, we rinsed the sample with water (Supplementary Movie 4). These hydrophobic properties are expected to enhance the long-term performance of the surface.

The results in Fig. 5 demonstrate that by altering the processing methods, the mechanical robustness of the microcavity is improved without significantly increasing the reflectance (Supplementary Fig. 21). The introduced silicon fabrication process and complementary polymer casting technique provides a platform for systematic engineering of multifunctional low reflectance materials tailored to application-specific requirements.

An important characteristic of the superblack microcavities is the low reflection for all incident angles. We demonstrate this property for our materials by bending a sample concavely and acquiring an image under intense illumination (~19,000 lx), Fig. 6a. We do the same when the sample is bent convexly, Fig. 6b. For both configurations the curvature of the sample is highly obscured, unlike previously demonstrated microcavity-based materials[1] and very little retroreflection is observed. Retroreflection may not be a significant drawback for some applications. However, reducing retroreflection is crucial in applications where preventing or complicating object identification with visible light is important, particularly when the object exhibits significant curvature or is viewed from glancing angles.

To demonstrate the wafer-scale uniformity and omni-angle low reflection capabilities of the fabricated superblack materials, we obscure the cylindrical geometry of a mug by covering it with the flexible material. A digital image of the mug without the superblack material is shown in Fig. 6c. Here, an illumination of ~19,000 lx is used. Due to the highly anisotropic reflectance profile of the mug surface, the shadings and highlights are clearly observed, allowing interpretation of the 3D geometry of the mug. Figure 6d is an image of the mug under the same illumination conditions, but when it is partially covered by the superblack material. Here, the shadings and highlights are

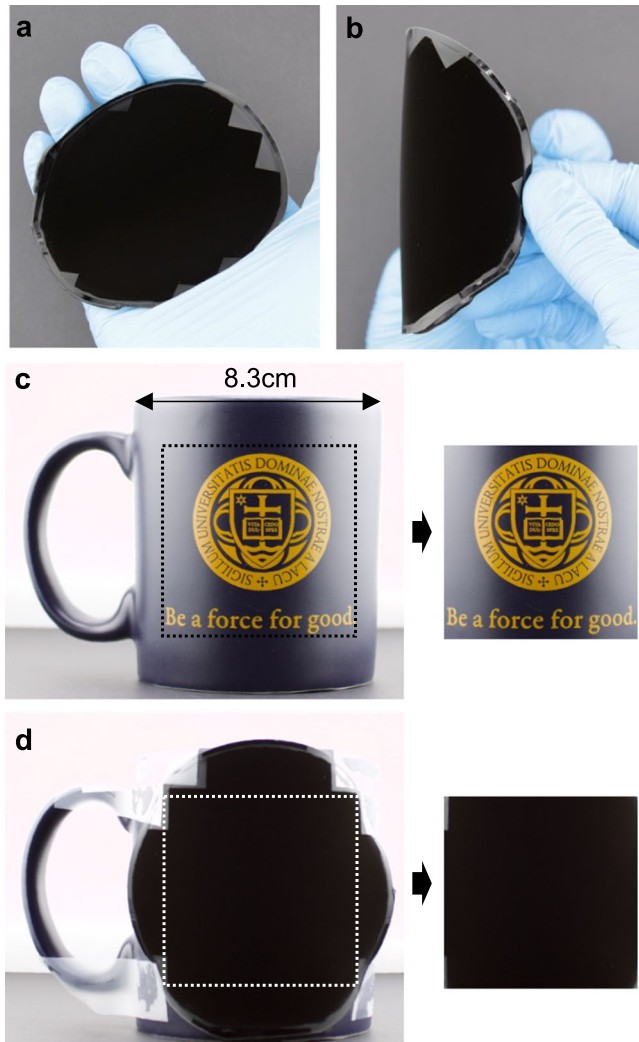

**Fig. 6 | Applications of the flexible superblack microcavity material.** Bending the flexible PDMS **a** concavely and **b** convexly under ~19,000 lx illumination inside of a light box. The sample shows no clear curve appearance. The shutter speed is 1/50 s for both photos. **c** A mug is taken under ~19,000 lx illumination where the surface curvature is clearly shown. **d** A mug is covered by H10G1 near flexible superblack microcavity material where no surface curvature can be seen. The shutter speed is 1/50 s for both images.

almost completely masked, making it difficult to interpret the 3D geometry of the mug.

## Discussion

In comparison to existing literature, we would like to summarize both the contributions and limitations of our final product from the perspective of microcavity design, fabrication, and performance and potential for scalability. Regarding microcavity design, fabrication, and performance, the key contributions include reducing retroreflection at glancing angles, enhancing surface robustness and hydrophobicity, and introducing more rigorous protocols for reflectance characterization. Also, in contrast to previous methods, our approach is deterministic and customizable, which enables the microcavity geometry to be rationally modified in future research and development. We acknowledge certain limitations of our material: our material shows higher $R_{Tot}$ at near-normal incidence compared to the ~0.02% reflectance reported in existing literature[1] and the range of wavelengths over which our material is superblack is in the visible

regime, which may limit some applications[1,2,4]. There are methods to address these limitations in the future. In particular, incorporation of different dyes and backing layers may reduce near-normal reflectance and improve the spectral range, similar to the previous work[1].

Regarding scalability, we demonstrated that the integrity of molds is maintained after repetitive casting and that multiple casts from the same mold display similar optical performance. While our current individual sample area is smaller than the state-of-the-art[1] by 30%, we demonstrated the viability of multiple castings to increase the overall textured area. Also, increasing the size of the silicon molds simply requires tools capable of processing larger wafers than those available at academic institutions. From a product manufacturing standpoint, microfabricated silicon molds provide advantages with respect to several critical considerations including mold size scalability (Supplementary Fig. 22), mold production costs (Supplementary Tables 4 and 5), mold throughput (Supplementary Table 4), production line ownership costs (Supplementary Table 5), supply chain logistics, and ease of integration. Without careful consideration of these aspects, the technology risks remaining limited to laboratory demonstrations. A detailed discussion about product manufacturing is provided in the Supplementary Discussion 1.

Finally, since there will likely be tradeoffs between optical performance (e.g., $R_{Tot}$ at normal incidence and the relative difference of $R_{Tot}$ at different angles), robustness, and manufacturability, the ideal microcavity design will likely vary for different applications, highlighting the importance of being able to deterministically modify microcavity design with the approach presented here.

In summary, we have introduced an engineering platform for flexible superblack materials utilizing standard silicon fabrication techniques and complementary PDMS casting methods. We have demonstrated potential manufacturing scalability by producing 4-inch wafer-scale silicon molds and microcavity samples, outlining a feasible path towards high-volume manufacturing of superblack materials. Rigorous reflectance measurements have been conducted to showcase its omni-angle superblack reflectance properties, which outperformed the current state-of-the-art. To demonstrate the application of this material, we have coated a 3D object with flexible superblack material, effectively concealing its 3D geometry. Furthermore, to illustrate the tunability of this platform, we have enhanced the mechanical robustness by modifying the etching process without significantly compromising its reflectance properties. Our engineering platform for superblack materials provides opportunities for the systematic engineering of future superblack materials.

## Methods

### Wafer-scale silicon pillar fabrication

The mask was manufactured using a direct laser writer (MLA150, Heidelberg Instruments) on a Chrome blank mask (Nanofilm). SPR700-1.2 photoresist (Rohm and Haas Electronic Materials) was spun at 4000 RPM on an HMDS-coated silicon wafer (University Wafer). Lithography was performed using i-line (GCA AutoStep 200 and Suss MicroTec MJB4 Mask Aligner) at a dose of 125 mJ cm$^{-2}$. DRIE (Alcatel A601E ICP-DRIE) etching process consists of etching and passivation steps. The etching step uses 150 sccm of sulfur hexafluoride ($SF_6$) at 42 mTorr pressure with 1600 W source power and 80 W plate power for 10 s with a bias voltage of 50 V. The passivation step uses 130 sccm Octa-fluorocyclobutane ($C_4F_8$) at 33 mTorr pressure with 1600 W source power and 40 W plate power for 2 s with a bias voltage of 60 V. The wafer is maintained at 20 ± 2 °C using backside liquid nitrogen cooling. The total DRIE etching step lasts 30 min. Oxygen plasma is then used to remove residue $C_4F_8$ and photoresist. The sharpening step is performed in the same DRIE at 60 mTorr pressure with 1600 W source power and 0 W plate power for various time: 10 min for 15 μm hexagon diameter, 7 min for 10 μm diameter, and 5 min for 7 μm diameter (or until the top surface is not observable by optical microscope, Olympus

MX61). Plasma ashing and RCA clean were performed to clean the wafer before oxidation. Wet oxidation step was performed in a diffusion furnace (Thermco Instrument Corp.) at 1150 °C for 2 h. Wafer was then placed in buffered HF (1:10) for 60 s.

### PDMS casting

The fabricated wafer is placed on a wafer carrier spider ring inside a desiccator (150 mm in diameter). A 60 mm crystallizing dish is placed under the divider with 1 mL of 98% TMCS (Sigma Aldrich). The desiccator is placed under vacuum inside a chemical hood for 20 min. A PDMS reagent solution (Sylgard 184 Silicone Elastomer, Dow) is mixed with Nigrosine (Sigma Aldrich) to produce an 8 wt% ratio of reagent to Nigrosine (placing Nigrosine first and reagent second in a vial). The mixture is sonicated using a 3 mm probe ultrasonic homogenizer (Bonvoisin) at the minimal power setting (12 W) for 30 s. Manual pulsing is required to prevent solution overheating. The resulting solution is mixed with PDMS base to achieve a 1 wt% Nigrosine mixture. The final solution consists of PDMS with a reagent to base ratio of 1:8 and 1 wt% Nigrosine. This mixture is then degassed for 20 min, or until no bubbles remain. A total of 8.5 grams of the final mixture is placed on the TMCS-coated wafer and subjected to a second 20-min degassing step. The wafer is then placed on a hotplate at 100 °C for 20 min. To ensure opacity, a second casting is performed on top of the first casting using 7 grams of 1 wt% Nigrosine PDMS mixture, following the same process. The resultant PDMS is carefully and slowly released from the wafer. Three consecutive castings were performed on the same wafer to produce Fig. 3c. 1 min of UV-Ozone treatment is performed within a few minutes after removal from the mold.

### Scanning electron microscopy imaging

All SEM images were captured using a field emission SEM (Helios 5 DualBeam, ThermoFisher Scientific) with an electron acceleration voltage of 10 kV and a probe current of 13 pA. The PDMS microcavity samples were coated with iridium to a thickness of ~10 nm using a DC magnetron sputtering coater with rotary-planetary tilting stages (TedPella). All measurements were conducted in standard mode, except for measurements for black silicon which were conducted in immersion mode.

### Coherent scanning interferometry

CSI measurements were conducted using a ZYGO NewView 9000 device equipped with a 50× Mirau objective lens. The vertical scan length was set to 145 μm using piezoelectric transducer. To maintain consistency in our measurements, we standardized the illumination intensity and shutter speed across all measurements. Due to the samples' low reflectance, we maintained maximum illumination intensity and using 15% shutter speed to prevent camera saturation, which would lead to data loss. Additionally, we enabled the signal oversampling option with 4× averaging settings to enhance the signal-to-noise ratio.

### Bidirectional reflectance distribution function

BRDF measurements were conducted using a custom-built goniometer controlled by a LabVIEW-programmed field programmable gate array (FPGA) controller. The goniometer adjusts the inner and outer arms independently to vary $\theta_i$ and $\theta_r$ angles, respectively. To minimize stray light reflections, the custom sample holder was coated with black paint. Optical power detection utilized a low-power silicon detector with a noise floor of 20 pW (Newport 918D-SL-OD3R), positioned 25 cm from the sample, corresponding to a solid angle of 0.0016 Sr$^{-1}$. Illumination was provided by a supercontinuum laser source (SuperK FIANIUM, Coherent) emitting at a wavelength centered at 532 nm with a 12 nm optical bandwidth. A collimating iris was placed in front of the laser, with post-iris power of 40 mW. Stringent alignment protocols ensured that the sample plane and laser co-aligned with the detector's center of rotation. A constant background signal was collected and subtracted from each measurement. The alignment protocols and experimental setup is further explained in the supplementary information. To validate the BRDF measurement accuracy, we estimated the total hemispherical reflectance using BRDF measurement at $\theta_i = 5°$ and $\theta_r = 0°$ to 80° assuming azimuthal symmetry around $\theta_r = 5°$ (supplementary materials). The total reflectance calculated from BRDF is nominally consistent with the integrating sphere measurement.

### Black silicon fabrication

A bare silicon wafer is placed in a DRIE system. The etching process consists of two steps: etching and passivation. During the etching step, 300 sccm of $SF_6$ is used at 60 mTorr pressure with 1800 W source power and 80 W plate power for 5 s. The passivation step follows, using 120 sccm of $C_4F_8$ at 30 mTorr pressure with 1800 W source power and 80 W plate power for 2 s. The total etching time is 20 min.

### Spectrally resolved integrating sphere measurement

All spectrally resolved total hemispherical reflectance measurements were performed using a double beam UV-Vis/NIR spectrometer with 60 mm integrating sphere accessories (Jasco V770). All measured samples were referenced to a 99% reflectance standard (Spectralon, Labsphere), calibrated traceably to the U.S. National Institute of Standards and Technology (NIST). The background was measured with the sample port open. To demonstrate measurement accuracy, we conducted the same procedure on a 2% reflectance standard (Spectralon, Labsphere) with NIST-traceable measurements performed by a third party, Labsphere. A comparison is provided in the supplementary material. The results indicate that our sample reflectance falls within either overestimating or being consistent with the standard.

### Tunable laser integrating sphere measurement

All laser based integrating sphere measurement with a sample side port mount is performed using 150 mm Reflectance/Transmittance spheres (RTC-060-SF, Labsphere). The integrating sphere is reconfigured to single-beam specular inclusive reflectance measurement. The illumination was provided by a supercontinuum laser source emitting at a wavelength centered at the specified wavelength with a 10 nm optical bandwidth. A high sensitivity silicon detector (SD-S1, Labsphere) with a resolution of 1 fW and a minimum power of 0.9 pW was used for power measurement. Both dark and baseline measurements were performed for every measured wavelength. Baseline measurement is reference to the same 99% standard as spectrally resolved integrating sphere measurement. Baseline to dark ratio is >10$^5$ (Supplementary Material).

### Angle dependent integrating sphere measurement

All angle dependent integrating sphere measurement is performed using 150 mm Reflectance/Transmittance spheres with a clip center mount[1], recommended for flexible materials (RTC-060-SF-CLIP, Labsphere). The illumination was provided by a supercontinuum laser source emitting at a wavelength centered at the specified wavelength with a 10 nm optical bandwidth. A ~1 × 4 cm$^2$ H15G1 is placed inside of a sphere using the clip mount resulting a sample area to sphere area ratio of <0.006, within the tolerance (<0.01) of ASTM E 903 section 7.2. One side is flat PDMS with Nigrosine and one side is casted H15G1 patterns. Both the dark and baseline measurements were performed while the sample is inside of the sphere (discussed in the Supplementary Materials). The resulted measurement at near normal incidence is consistent with the side port mount.

### Photography and video recordings

The lightbox photos were taken inside a 16 inches × 16 inches × 16 inches studio lightbox with all 480 LEDs on (Puluz). The illumination

intensity was measured using a luxmeter (MT-912 URCERI). Images within the lightbox were captured using a digital single-lens reflex camera (Canon EOS Rebel T7 with EF-S 18-55 mm lens) at ISO 100 in manual mode. The aperture size was kept constant for comparative images. Color balance was adjusted using optical whites under 21,000 lx illumination inside the lightbox, at the maximum shutter speed before saturation. No further modifications were made to the images. Figure 4e, f, the camera was mounted on a tripod (K&F Concept) to ensure accurate camera angles. The setup shown in Fig. 3b was captured using an iPhone 7 Plus (Apple) in automatic mode within a chemical hood. The wafers were placed on top of cleanroom wipes. All the videos were recorded using an iPhone 7 Plus (Apple) in automatic mode.

## Surface testing

Approximately 2 Newtons of normal load is applied to the sample surface, measured using a digital scale (VWR), through a curved-head metal tweezer with a contact area of ~1 × 3 mm$^2$, resulting in a pressure of ~600 kPa, as shown in the Supplementary Information. The sample is placed in water for 5 min, then removed and blasted with ~100 kPa static pressure dry air for more than 1 min. To further increase the air velocity, we put a micropipette at the air exit. Ultrasonic cleaner is purchased from VWR Symphony. For the R-H15G1 surface testing, we performed surface scratching using the same curved-head metal tweezer while applying 4 different normal loads: ~150 kPa, ~300 kPa, ~600 kPa, and ~900 kPa. We further performed Grit 2000 sandpaper scratching test using ~300 kPa. The finger touching test assumes a clean finger with a contact surface area of 1 × 2 cm$^2$. The scotch roller duster is from Scotch-Brite Everyday Clean Lint Roller. We conducted contact angle measurements using a DataPhysics OCA device.

## Data availability

The main text experimental data generated in this study are provided in the Supplementary Information/Source Data file. All data are available from the corresponding author upon request. Source data are provided with this paper.

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

## Acknowledgements

The authors gratefully acknowledge the funding support for this research from the United States government. The authors acknowledge financial support to Y.Y. from the University of Notre Dame through the Interdisciplinary Materials Science and Engineering Fellowship. The authors thank Notre Dame nanofabrication facility for the use of the cleanroom and all the mentioned silicon fabrication equipment. SEM imaging was carried out in part in the Notre Dame Integrated Imaging Facility, University of Notre Dame using Helios scanning electron microscope. The authors thank the ND Energy Materials Characterization Facility (MCF) for the use of the UV-Vis spectroscopy. The MCF is funded by the Sustainable Energy Initiative, which is part of the Center for Sustainable Energy at Notre Dame.

## Author contributions

Y.Y. and M.R.R. conceived the study. Y.Y. developed the silicon mold fabrication process. Y.Y. and M.R.R. developed the PDMS casting process. Y.Y. performed the SEM imaging and took lightbox photos. Y.Y. set up and conducted the integrating sphere measurements. B.S., Y.Y., A.J.H., A.P., and E.C.K. set up and performed the BRDF measurements. Y.Y. carried out surface robustness testing. B.S. performed hydrophobicity and CSI measurements. Y.Y. drafted the manuscript. M.R.R., A.J.H., and Y.Y. edited the manuscript. All authors contributed to the discussion of the results. M.R.R., A.J.H., and E.C.K. supervised the project.

## Competing interests

Y.Y., M.R.R., A.J.H., and E.C.K. declare the following competing interest: Y.Y., M.R.R., A.J.H., and E.C.K. have submitted a provisional patent (application # 63/716,925) through the University of Notre Dame about the design and production of superblack materials described in this manuscript. Y.Y., M.R.R., A.J.H., and E.C.K. declare no other competing interests. B.S. and A.P. declare no competing interests.
