## [Transparent Peer Review file · Nature Communications]

Engineering Flexible Superblack Materials

Corresponding Author: Professor Matthew Rosenberger

Parts of this Peer Review File have been redacted as indicated to remove third-party material and confidential information.

Version 0:

Reviewer comments:

Reviewer #1

(Remarks to the Author)

In this manuscript, the authors reported the preparation of flexible superblack PDMS films using a template method. The resultant films are broadband and omni-angle superblack, and scratching resistant. The properties of the films can be tuned by using molds with different microstructures. Although the templating method is not new, and it is predictable that a flexible superblack surfaces could be obtained by using elastomers to form the microstructures, this manuscript still deserves publish in Nat. Commun. due to the excellent performance of the final product. Followings are suggestions for the authors:

1. Different from other superblack materials based on carbon, semiconductor and metal materials, PDMS is an elastomer. The authors should mention this point when discussing the flexibility and scratching resistance performances in the text. Since the authors have black silica, a typical fragile superblack material at hand, it is suggested to use black silica as the control sample to perform the tweenzer pressing and sandpaper scratching investigations. The comparison of the results will be convincing to demonstrate the advantages of PDMS superblack films.
2. It is not suitable to declare that "PDMS is then released from the microcavity mold without leaving a residue". SEM images only show a very limited area and the images can also not identify the little residue left. The mold can be reused several times even though there were small amount of PDMS left or the microstructure of the mold was destroyed a little.
3. The authors should check the description "Under intense illumination of ~19,000 lux, and 1/50 s shutter speed, there is no strong reflectance variation between consecutive castings (Figure 3c). Under this illumination and shutter speed, a reflectance variation of >0.1 % is observable (Figure S2) suggesting no clear degradation of the silicon molds after three consecutive castings." In Figure S2, the speed is 1/200s, and illumination of 21,000 lux; and it might be <0.1 %, not >0.1%.
4. Why the authors did not offer the sideview SEM images of the PDMS films in the main text or SI? This information is very important and can not be neglected.
5. How about the hydrophobicity of the film? Maybe it is superhydrophobic because of the microstructures on the surface and the superhydrophobicity is benefit for its longtime performance. Experiment of wetting behavior is suggested to be added.

Reviewer #2

(Remarks to the Author)

This manuscript reports a new technique for fabricating superblack materials by transferring microcavity structures onto black PDMS, based on templates of microcavity patterns fabricated by the deep RIE process on silicon substrates. Optimized microcavity structure patterns efficiently confine incident light and prevent reflections, reducing net reflectance to a minimum of 0.15 % in the visible wavelength range. The resulting superblack material is durable and can be produced repeatedly, and is therefore recognized as an achievement that expands the range of applications of superblack materials. The manuscript is well written. The methodology is also well described with sufficient detail of both production and evaluation methods, and no fatal flaws are found. Most of the claims are supported by data.

Throughout the manuscript, the superiority of the various properties of the developed superblack material is discussed against Ref. 1 as a benchmark. Ref. 1 is a reasonable comparison since the superblack material in this study is similar in concept (microcavity) and fabrication method (mold replication) to that described in Ref. 1. Certainly, there is nothing wrong with the claims made in this paper that the superblack material of this study has higher durability, weaker dependence on angle of incidence and lower retroreflection. However, a crucial difference between this study and Ref. 1, which is not mentioned much in the present manuscript, is that the minimum reflectance at near normal incidence is still an order of magnitude higher. In addition, the absolute value of reflectance at a large angle of incidence (at 75 degrees) is almost the

same (~2 %) between this study and Ref. 1. The reason why the superblack in this study appears to have a relatively weak dependence on angle of incidence is that the minimum reflectance at near normal incidence is an order of magnitude higher than in Ref. 1 (~0.02 %). Some discussion of the limitations and scope for improvement of the present study should also be made. This work would remain valuable even if the minimum reflectance were limited to the values reported here. In this case, however, the comparison of durability and/or angle of incidence properties should include not only Ref. 1, but also Ref. 2, Ref. 4 and/or commercial superblacks* with similar minimum reflectance values to those in this study.

*For example, <https://the-black-market.com/>

In short, while I appreciate the effort to develop a novel superblack material in this study, I feel that this manuscript needs to address the above concerns to justify its publication in this journal. In addition, I would like to make the following specific comments.

Specific comments:

- Introduction, 8th line from the end of the last paragraph, and p.8 third sentence: As noted in the main concern above. The claim of weak dependence on the angle of incidence here is somewhat misleading.
- Ref. 12: The literature information is insufficient.
- Figure 3(c): On the PDMS replica, the area corresponding to the flat part of the Si mold appears transparent. Why is this?
- P.8, first sentence: What does "outperform" mean? What are the advantages over Lambertian characteristics?
- Figure S12c: It is strange to assume $n = 0.93$, which is not possible for natural materials.
- The SI unit of illuminance is lx, not lux.
- Are the results in Figure S18 obtained after the sandpaper scratch test? If so, this should be stated in the figure legend.

Reviewer #3

(Remarks to the Author)

This manuscript reported flexible and durable supreme black materials based on microcavities of PDMS mixed with nigrosine. Lambertian characteristics with a minimum reflectance of 0.15% and less than 0.4% across the visible spectrum were obtained. Certain durability tests were carried out. Visual demonstrations under bright illumination were also carried out to showcase the uniformity. Below are my comments that need to be addressed prior to making a decision.

1. The main target of this manuscript is to solve two potential issues in Amemiya et al, Sci. Adv., 2023, 9, ade4853 (ref [1]): the first one is the retroreflection and the second one is the mold longevity and scalability. For the first issue, the current manuscript still has strong retroreflection at normal angles. Moreover, the reflection at higher angles of incidence is not really a "significant improvement over state-of-art [1]". Ref. [1] has a much broader supreme black range. For the second issue, quantitative data on mold longevity should be provided for the method reported in this manuscript. The size of a single piece is in fact smaller than that shown in Ref. [1]. The contribution of this manuscript should be more rigorously defined.
2. Retroreflection is not necessarily a drawback for most applications of supreme black materials, such as straylight. The authors should articulate why retroreflection needs to be targeted.
3. Also, for most applications of supreme black materials, low reflectance over the entire UV, visible and near infrared range is required. The authors should discuss the applicability of the current method in broadband black materials.
4. The durability test is insufficient. The high pressure in this manuscript is due to the small area of the tweezer tip. The impact area is too small to assess the durability. Finger touch or roller duster test should be added along with reflectance spectra before and after the durability test.

Version 1:

Reviewer comments:

Reviewer #1

(Remarks to the Author)

The authors have thoroughly addressed my concerns, and the revised manuscript is now suitable for publication.

Reviewer #2

(Remarks to the Author)

I appreciate that the authors have carefully and thoroughly revised the manuscript. Now I believe the reviewers' comments have been adequately addressed. I am pleased to recommend this work for publication in Nature Communications as it is.

Reviewer #3

(Remarks to the Author)

The authors have properly revised the manuscript. I recommend the publication of this paper.

Reviewer #1 (Remarks to the Author):

In this manuscript, the authors reported the preparation of flexible superblack PDMS films using a template method. The resultant films are broadband and omni-angle superblack, and scratching resistant. The properties of the films can be tuned by using molds with different microstructures. Although the templating method is not new, and it is predictable that a flexible superblack surfaces could be obtained by using elastomers to form the microstructures, this manuscript still deserves publish in Nat. Commun. due to the excellent performance of the final product. Followings are suggestions for the authors:

1. Different from other superblack materials based on carbon, semiconductor and metal materials, PDMS is an elastomer. The authors should mention this point when discussing the flexibility and scratching resistance performances in the text. Since the authors have black silica, a typical fragile superblack material at hand, it is suggested to use black silica as the control sample to perform the tweezer pressing and sandpaper scratching investigations. The comparison of the results will be convincing to demonstrate the advantages of PDMS superblack films.

Response:

We thank the reviewer for suggesting including the advantage of PDMS as an elastomer and a comparison of scratch resistivity between black silicon and PDMS. This method effectively differentiates the scratch resistivity of RH15G1 from that of existing materials. We conducted a scratch test on both black silicon and RH15G1, as shown in Figure R1.

Figure R1. Scratch test on black silicon and RH15G1. The top image shows the black silicon after the scratch test, while the bottom image shows RH15G1 after the test. (A video is attached for verification.)

Changes made in response to comment:

1. Included a statement about elastomers.
2. Included the scratch test video to the supporting information.
3. Included Figure R1 to the main text (Figure 5j,k) and discussion about black silicon after scratching is also included in the maintext.

2. It is not suitable to declare that “PDMS is then released from the microcavity mold without leaving a residue”. SEM images only show a very limited area and the images can also not identify the little residue left. The mold can be reused several times even though there were small amount of PDMS left or the microstructure of the mold was destroyed a little.

Response:

We thank the reviewer for pointing out the lack of rigor in the original statement. We agree with the reviewer and have performed additional SEM imaging of larger areas. The image below shows the mold after undergoing more than five casts. The limited imaging area is due to the field of view of the SEM system. We would also like to note that no iridium or gold was sputtered onto the mold. Therefore, if there were any PDMS residue, significant charging would be observed. We still agree with the reviewer that “without leaving a residue” is a very strong statement and change to “We observed no residue using SEM imaging in an area $>4 \text{ mm}^2$.”

2 mm

Figure R2. SEM images of the mold after more than five casts, taken at different magnifications to verify the presence of any PDMS residue.

Changes made in response to comment:

1. Including the SEM images in the Figure R2 to the supporting information.
2. Revising the original text “without leaving a residue” to “without leaving a significant amount of residue.”
3. We replaced the original text that support the argument “without leaving a significant amount of residue” to “Based on the SEM inspection in a more than 4 mm² area region, we found no significant PDMS residues after more than five casts shown in Figure S2.”

3. The authors should check the description “Under intense illumination of ~19,000 lux, and 1/50 s shutter speed, there is no strong reflectance variation between consecutive castings (Figure 3c). Under this illumination and shutter speed, a reflectance variation of >0.1 % is observable (Figure S2) suggesting no clear degradation of the silicon molds after three consecutive castings.” In Figure S2, the speed is 1/200s, and illumination of 21,000 lux; and it might be <0.1 %, not >0.1%.

Response:

We thank the reviewer for pointing out the confusion. We would like to clarify that Figure S2 includes images taken at different shutter speeds. The third panel from the top was captured with a shutter speed of 1/50s and an illumination of 21,000 lx (which fluctuates between 20,000 lx and 21,000 lx). The variation in illumination is primarily due to the light source and whether the front reflective panel of the light box is enclosed. When the front panel was not enclosed, the illumination to range between 19,000 lx and 20,000 lx which was the illumination condition of Figure 3c. However, we do not believe these small variations in illumination impact the original statement. We agree with the reviewer that a reflectance variation of less than 0.1% can often be observed. However, to be conservative and hopefully more clear, we have modified the statement to be “...approximately 0.1% or more...”, as it provides a margin for error in this semi-qualitative measurement.

During the revision, we have conducted additional quantitative measurements of multiple castings of the same mold and found that the total range of values between five different casts is ~0.03%, as shown in Figure R3. This measurement provides a quantitative metric for the repeatability of our approach. We have included these results in Figure S3 in the revised supplementary information.

Figure R3. Quantitative comparison of total reflectance at near-normal incidence angle for 5 casts of the same R-H15G1 mold.

Changes made in response to comment:

1. We added a statement in Figure S2 figure description: “The shutter speed for the top panel to bottom panel varied as follows: 1/200s, 1/100s, 1/50s, and 1/25s.”
2. We modified the statement about Figure 3c to be “...approximately 0.1% or more...” to be more clear and maintain a conservative estimate.
3. We added a sentence to the paragraph describing Figure 3c about our quantitative analysis of casting repeatability.
4. We added Figure S3, which summarizes the quantitative repeatability measurement.

4. Why the authors did not offer the sideview SEM images of the PDMS films in the main text or SI? This information is very important and can not be neglected.

Response:

We thank the reviewer to point out that the sideview or the cross-sectional view of the PDMS was not included in the original manuscript. We agree with the reviewer that cross-sectional view of the PDMS is essential and cannot be neglected, especially when arguing an improvement in the microcavity design and fabrication. Thus, we took cross-sectional view of the PDMS microcavity and measure the entrance slope / aspect ratio (Figure R4).

Figure R4. Cross-Sectional view of the H10G1 microcavity. The entrance slope is measured to be $\sim 4^\circ$ or an aspect ratio of $>13:1$.

Changes made in response to comment:

1. We replaced the original Figure 3g with Figure R4 to demonstrate the high aspect ratio at the microcavity entrance.
2. We discussed the cross-sectional view of the H10G1 microcavity in the main text: “At the entrance of the microcavity, the SE is measured to be $\sim 4^\circ$, resulting in an aspect ratio of >13 , which exceeds the reported value of ≥ 4 or >5 in the existing literature. Steep S_E should result in better glancing angle performance.”

5. How about the hydrophobicity of the film? Maybe it is superhydrophobic because of the microstructures on the surface and the superhydrophobicity is benefit for its longtime performance. Experiment of wetting behavior is suggested to be added.

Response:

We appreciate the reviewer’s comments regarding the hydrophobicity of the surface and agree that demonstrating these properties would enhance the performance evaluation of the product. To assess the hydrophobic nature, we conducted contact angle measurements using a DataPhysics OCA device. Six individual measurements were taken at different locations on the sample. The average contact angle was determined to be 137.6° with a standard deviation of 3.6° . The individual measurements are as follows: 136.3° , 133.8° , 141.2° , 132.8° , 140.7° , 141.7° , and 136.7° . Figure R5a illustrates a $3\ \mu\text{L}$ droplet resting on the surface of H10G1. Additionally, to further validate

the hydrophobic properties, we rinsed the sample with water, as shown in Figure R5b. A video of the water rinse test is also provided for reference.

Figure R5. Hydrophobicity characterization of the superblack material: (a) Contact angle measurement, with the measured contact angle of 141.7°. (b) Water rinsing test demonstrating the hydrophobic behavior of the surface.

Changes made in response to comment:

1. We included contact angle measurement and water rinsing test in Figure 5 and updated the caption for Figure 5.
2. We included a summary of the above response in the paragraph describing Figure 5m and 5n.
3. We included the contact angle measurement procedure in the methods section.
4. The video of the water rinsing test is included in the supporting information.

Reviewer #2 (Remarks to the Author):

This manuscript reports a new technique for fabricating superblack materials by transferring microcavity structures onto black PDMS, based on templates of microcavity patterns fabricated by the deep RIE process on silicon substrates. Optimized microcavity structure patterns efficiently confine incident light and prevent reflections, reducing net reflectance to a minimum of 0.15 % in the visible wavelength range. The resulting superblack material is durable and can be produced repeatedly, and is therefore recognized as an achievement that expands the range of applications of superblack materials.

The manuscript is well written. The methodology is also well described with sufficient detail of both production and evaluation methods, and no fatal flaws are found. Most of the claims are supported by data.

Throughout the manuscript, the superiority of the various properties of the developed superblack material is discussed against Ref. 1 as a benchmark. Ref. 1 is a reasonable comparison since the superblack material in this study is similar in concept (microcavity) and fabrication method (mold replication) to that described in Ref. 1. Certainly, there is nothing wrong with the claims made in this paper that the superblack material of this study has higher durability, weaker dependence on angle of incidence and lower retroreflection.

1. However, a crucial difference between this study and Ref. 1, which is not mentioned much in the present manuscript, is that the minimum reflectance at near normal incidence is still an order of magnitude higher. In addition, the absolute value of reflectance at a large angle of incidence (at 75 degrees) is almost the same (~2 %) between this study and Ref. 1. The reason why the superblack in this study appears to have a relatively weak dependence on angle of incidence is that the minimum reflectance at near normal incidence is an order of magnitude higher than in Ref. 1 (~0.02 %). Some discussion of the limitations and scope for improvement of the present study should also be made. This work would remain valuable even if the minimum reflectance were limited to the values reported here.

Response:

We would like to thank the reviewer for pointing out that the original manuscript did not clearly explain the cause of the lower retroreflectance and the weaker dependence on the angle of incidence. This led to the interpretation that the reduced angle-dependence was a result of higher reflectance at near-normal incidence. To clarify, we believe that the reduced angle dependence and weaker retroreflection is related to the cavity geometry, namely the entrance slope as defined in Figure 1. This is because a steeper entrance slope will reflect light rays from large angles deeper into the cavity where they can reflect several times before exiting the cavity. In Ref. 1, the entrance aspect ratio is stated as ≥ 4 , which corresponds to an entrance slope of 14.0° . Based on the cross-sectional view of the microcavity in our work (Figure R6), the entrance aspect ratio is approximately 13, corresponding to an entrance slope of 4.3° . Therefore, we believe that the observed lower retroreflectance and weaker angular dependence are likely due

to superior microcavity entrance fabrication techniques in this work, which improve the angle-dependent performance relative to Ref. 1.

We agree with the reviewer that the original text did not clearly state that the total reflectance at near-normal incidence is still close to an order of magnitude greater than the Ref. 1. We will clearly state this limitation of the presented material. We are not sure why the material in Ref. 1 has this performance enhancement at near-normal incidence. We speculate that the special cashew nut shell liquid used in Ref. 1 could further reduce our reflectance values. However, Cashew Nut Shell Liquid (No. 91) used in Ref. 1 is not listed in the U.S. Environmental Protection Agency's Toxic Substances Control Act (TSCA) chemical substance inventory. As a result, CASHEW COMPANY LTD. is unable to export this chemical from Japan to the United States (see email in response to Reviewer 3).

Figure R6. Cross-Sectional view of the H10G1 microcavity. The entrance slope is measured to be $\sim 4^\circ$ or an aspect ratio of $>13:1$.

Changes made in response to comment:

1. We replaced the original Figure 3g with Figure R6 to demonstrate the high aspect ratio at the microcavity entrance.
2. We explicitly stated the limitation of the material at the near normal incidence angle which is still close to one magnitude higher than what was reported in Ref. 1 in the main text.
3. We explicitly stated the scope of the improvement is limited to the relative different between the near normal incidence. However, we suspect that the

improvement is due to the higher entrance slope of the microcavity in comparison with Ref. 1.

2. In this case, however, the comparison of durability and/or angle of incidence properties should include not only Ref. 1, but also Ref. 2, Ref. 4 and/or commercial superblacks* with similar minimum reflectance values to those in this study.

*For example, <https://the-black-market.com/>

Response:

We would like to thank the reviewer for the valuable suggestion to compare our results with additional existing literature, beyond Ref. 1, as well as with commercially available superblacks. We included near normal incidence total integrating sphere comparison with Ref. 2. and Ref. 4. We included durability comparison with Ref. 4. However, Ref. 2. and Ref. 4 did not report angle dependent total integrating sphere measurement. The angle dependent specular measurement reported in Ref. 4 is similar to the BRDF measurement reported in this manuscript, but the measurement setup is very different. To compare, we included the raw BRDF data in Watts captured by our detector at 40 mW 532 nm laser illumination. This will allow the readers to extract data from the raw measurement. We have several commercially available superblack materials, as shown in Figure R7. However, due to legal and confidentiality constraints, we are unable to directly compare our sample with these commercial products in this manuscript.

We will also add that the Vantablack S-Vis reflectance data is publicly available but exhibits significant inconsistency, even within the same document and across different sources (Figure R8). As such, comparisons between samples based on publicly available data can be challenging. This is the primary reason we rigorously characterized the reflectance of the reported sample in this manuscript, employing multiple comparisons to ensure consistency:

1. Reproducing third-party calibrated reflectance measurements (Figure S10).
2. Ensuring consistency across two distinct reflectance measurement systems, using different light sources, detectors, and setups (Figure 4a).
3. Comparing BRDF measurements with total hemispherical reflectance at near-normal incidence (Figure S11).

For completeness, we also include other publicly available data for commercially available low-reflectance materials in Figure R9. However, the reported reflectance values for these materials are higher than those presented in this manuscript, with the exception of Musou Black Fabric KIWAMI at near-normal incidence. Unfortunately, angle-dependent total hemispherical reflectance data for Musou Black Fabric KIWAMI is not publicly available. Furthermore, due to the flocking process and flammability concerns noted in Ref. 4, the potential applications of Musou Black Fabric KIWAMI may be limited.

[REDACTED]

Figure R7. Commercially Available Low Reflectance Materials: The two small boxes in the top-left corner represent S-Vis and S-IR from Vantablack. The middle two containers contain Musou Black and Black 2.0. Vantablack VBx2 and flocked fabric are labeled as shown on the packaging.

[REDACTED]

Figure R8. Inconsistent Publicly Available Data for Vantablack S-Vis: (a) and (b) are from the same document, reporting total hemispherical reflectance values of 0.036% and 0.1% at 700 nm, respectively. (c) presents data from a different document, which states a total hemispherical reflectance greater than 0.2%. The difference between the lowest and highest reflectance measurements from the same company spans nearly an order of magnitude.

[REDACTED]

Figure R9: Reflectance Data for Other Commercially Available Low Reflectance Materials: Panels (c) and (d) show the reflectance data for Vantablack VBx2.

Changes made in response to comment:

1. We included comparisons with Ref. 2. and Ref. 4. in addition to Ref. 1 for the mechanical tests. Also, we included Ref. 2. and Ref. 4 regarding the limited spectrum performance.
2. The limitation of the presented material and outlook for further improvement is included in the end of the manuscript in a section titled "Discussion and Outlook".

In short, while I appreciate the effort to develop a novel superblack material in this study, I feel that this manuscript needs to address the above concerns to justify its publication in this journal. In addition, I would like to make the following specific comments.

Specific comments:

- Introduction, 8th line from the end of the last paragraph, and p.8 third sentence: As noted in the main concern above. The claim of weak dependence on the angle of incidence here is somewhat misleading.

Response:

We agree with the reviewer and removed the statement.

- Ref. 12: The literature information is insufficient.

Response:

We agree with the reviewer and will include more information in the citation.

- Figure 3(c): On the PDMS replica, the area corresponding to the flat part of the Si mold appears transparent. Why is this?

Response:

The region on the PDMS replica corresponding to the flat part of the Si mold exhibits specular reflection and the bright spots are specular reflection from lights above the sample. The attached image shows that the mold is not transparent at the flat region.

Figure R10. PDMS superblack on a transparent plastic petri dish.

- P.8, first sentence: What does “outperform” mean? What are the advantages over Lambertian characteristics?

Response:

We agree with the reviewer that the original wording was unclear. The superblack material described is a Lambertian material with 0.4% reflectance. To avoid confusion, we have

removed "Lambertian" from the original sentence and rewrite the sentence to be more clear.

Changes made in response to comment:

We restated "Under 532 nm laser illumination at $\theta_i = 5^\circ$, the lowest reflectance microcavity sample, H15G1, exhibiting reflectance $< 0.4\%$ for θ_r ranging from -70° to 70° (red dashed line in Figure 4b)."

- Figure S12c: It is strange to assume $n = 0.93$, which is not possible for natural materials.

Response:

We agree with the reviewer that $n = 0.93$ is not possible for natural materials for broadband spectrum and is only possible for a narrow spectral range using metamaterials. To improve the generality, we did not limit this calculation to the natural materials. We are happy to remove the figure if the reviewer thinks that this figure is confusing.

- The SI unit of illuminance is lx, not lux.

Response:

We thank the reviewer for catching this mistake.

Changes made in response to comment:

We have change all "lux" to "lx".

- Are the results in Figure S18 obtained after the sandpaper scratch test? If so, this should be stated in the figure legend.

Response:

Figure S19 (original Figure S18) is not obtained after the sandpaper scratch test. Figure S18 contains measurement of RH15 series with varying G.

Changes made in response to comment:

We included reflectance comparison for pristine, finger touch, roller duster and Tweezer scratch in Figure 5.

Reviewer #3 (Remarks to the Author):

This manuscript reported flexible and durable supreme black materials based on microcavities of PDMS mixed with nigrosine. Lambertian characteristics with a minimum reflectance of 0.15% and less than 0.4% across the visible spectrum were obtained. Certain durability tests were carried out. Visual demonstrations under bright illumination were also carried out to showcase the uniformity. Below are my comments that need to be addressed prior to making a decision.

1. The main target of this manuscript is to solve two potential issues in Amemiya et al, Sci. Adv., 2023, 9, ade4853 (ref [1]): the first one is the retroreflection and the second one is the mold longevity and scalability. For the first issue, the current manuscript still has strong retroreflection at normal angles. Moreover, the reflection at higher angles of incidence is not really a “significant improvement over state-of-art [1]”. Ref. [1] has a much broader supreme black range. For the second issue, quantitative data on mold longevity should be provided for the method reported in this manuscript. The size of a single piece is in fact smaller than that shown in Ref. [1]. The contribution of this manuscript should be more rigorously defined.

Response:

We sincerely appreciate the reviewer's thoughtful comments and questions regarding the contribution of this manuscript. We would like to first acknowledge that Ref. 1 does indeed exhibit lower absolute sample reflectance, as noted by the reviewer. Ref. 1 has a much broader supreme black range and demonstrated a ~30% increase in individual sample size in comparison with current manuscript, as the reviewer correctly pointed out. We will change the statement in the main text to address these drawbacks.

Below, we wish to clarify the contributions of our work, both in terms of microcavity design, fabrication, and performance and potential for scalability.

With regard to microcavity design, fabrication, and performance, our work provides several key contributions: reduced retroreflection at glancing angles due to steep entrance slope, deterministic and customizable fabrication strategy for microcavities, improved surface robustness and hydrophobicity, and introduction of more rigorous protocols for reflectance characterization. Specific points are address below:

1. We created microcavities with the steepest reported entrance aspect ratio in the literature for this type of structure. The entrance aspect ratio in our samples is ~13 (see Figure 3g in updated manuscript or Figure R6 above), compared to the ≥ 4 reported in Ref. 1. We hypothesize that this improvement plays a key role in reducing retroreflection at glancing angles, a factor that is important for applications such as camouflaging spatial curvature.
2. With regard to the significant difference between the near-normal-incidence total reflectance between Ref. 1 and our work, we believe a significant part of the improved performance in Ref. 1 is due to the scattering properties of the cashew nut shell liquid. We attempted to purchase this product in order to isolate the

impact of microcavity geometry (focus of our work), but we were unable to purchase it due to export restrictions (see email later in response).

3. We present a more rigorous and complete approach to reflectance characterization, incorporating Bidirectional Reflectance Distribution Function (BRDF) analysis (Figure 4d), a comparison of reflectance measurements between two distinct systems (Figure 4a), and an accuracy comparison of integrating sphere measurements with third-party data (Figure S10). While the detector noise was characterized in Ref. 1, these additional reflectance techniques were not included. The inclusion of BRDF analysis allows for the observation of approximately 0.4% specular reflectance from the bottom of the PDMS microcavity, an insight that would not be possible without BRDF measurements, which were not included in Ref. 1. Further, even for vertical aligned carbon nanotubes and black silicon, the specular reflectance at near normal incidence is also observed in BRDF, presented in Ref. 6 and Ref. 7, so we do not believe this is a particularly deficiency of our material.
4. We developed a deterministic and customizable approach for the creation of microcavity geometries with sub-micron precision and repeatability. The use of standard cleanroom fabrication processes, namely photolithography, etching, and oxidation, to make molds provides a large degree of tunability of both the planar geometry of the cavities through mask design and the three-dimensional profile of the cavity through etch and oxidation processes. For different end applications, there will likely be tradeoffs between different material performance metrics, such as total reflectance at normal incidence, angle-dependence, and robustness. Our general approach of using microfabricated silicon molds allows for rationale design to optimize amongst such tradeoffs.
5. As one example of the flexibility of our approach, we modified our etch recipe to change the cavity entrance to improve robustness while maintaining excellent optical performance. We have conducted additional robustness testing in response to the reviewer's comments (presented later) that show that our materials can withstand tweezer scratches and direct contact with bare fingers, with stress resistance exceeding five times that reported in Ref. 1. Additionally, further demonstrations of improved hydrophobicity are presented in this updated manuscript.

With regard to mold longevity, we present evidence (much of it acquired as part of this revision) that our molds can be used for many casts without significant degradation of performance. Specific points are addressed below:

1. As part of the revision, we imaged one of our molds using SEM after more than five casting cycles and observed no evidence of PDMS residue over an area greater than 4 mm². This demonstration of mold integrity and repeatability was not presented in Ref. 1. These data are included in the revised supplementary information, Figure S1.
2. Additionally, we have measured the near-normal-incidence total reflectance from several castings from the same mold and found only approximately +/- 10% variation from the nominal value, further demonstrating the durability of our molds during repeated castings. These data are in Figure S3 of the revised manuscript).

3. While the individual casts presented in this manuscript are smaller than those shown in Ref. 1, the total superblack area demonstrated is larger. We provided seven casts of 4-inch scale PDMS superblack samples shown in Figure R11. It is important to note that we have sent PDMS superblack to external stakeholders for verification and exhibition. Figure R11 represent only the PDMS superblack that remain with us. With silicon fabrication, the individual mold area can be further expanded, as will be discussed later.

Figure R11. Presenting seven 4 inch superblack materials

From a product manufacturing perspective, we wish to address key factors such as wafer size scalability, mold production costs, mold throughput, production line ownership costs, supply chain logistics, and ease of integration. These elements are critical for translating scientific research into commercially viable products. Without considering these factors, the technology is likely to remain confined to laboratory demonstrations rather than reaching full commercial applicability.

1. Silicon fabrication-based processes offer the potential to scale mold production from 100 mm to ≥ 200 mm diameter wafers, as 200 mm and 300 mm wafer production lines are common in industry-scale manufacturing. Currently, the limitation in wafer size is due to the constraints of the tools used for basic science research institutions. In this work, we successfully scaled from a 75 mm diameter pattern to a 100 mm diameter mold with minimal investment in research and process development (Figure R12). Additionally, we are collaborating with external contractors to scale production from 100 mm to 150 mm diameter molds. While further process development will be required to scale production from 100 mm to 150 mm wafers, no additional investments in research are needed.

Figure R12. Demonstrating potential for scaling from 75 mm to 100 mm.

2. While the authors in Ref. 1 suggest that master mold replication could reduce costs, they did not demonstrate the feasibility of repetitive replication. In contrast, our updated manuscript includes a cost analysis based on current labor market rates and process cost estimations (Table R1). The conservative cost for fabricating an original silicon mold is estimated at under \$500, with a mold throughput of less than 50 minutes per mold. This throughput can potentially be further improved by optimizing the DRIE etching parameters. These cost and throughput analyses were not presented in Ref. 1.

Table R1. Cost and throughput analysis for silicon mold production.

	Time (minutes)	Cost per wafer (\$)	Details
Skilled Labor	99.2	79.5	
Cleanroom Access	99.2	165.3	
Photoresist deposition	1	1.59	2mL
Lithography Tool	0.6	1.71	0.6 minutes
DRIE	45	85.6	45 minutes
Oxidation	3	0.04	300 minutes per 100 wafers
Bare Silicon Wafer	0	13.46	100 mm wafer
	Total Cost (\$)	347.2	
	Throughput per Wafer (minutes)	49.6	Limited by number of DRIE

	Cost (\$)	Unit	Cost	Unit	throughput	additional details
Skilled Labor	48.1	per hour	\$100,000	per year		
Cleanroom Access	100	per hour				Cleanroom Access Includes basic chemicals
Photoresist	0.79	per mL	\$3,000	per Gallon		
Lithography Tool	2.85	per minutes	\$1,500,000	per Tool	100 wafers per hour through put	Referbished i-line stepper, Cost per minutes is based 1 year of use
DRIE Tool	1.90	per minutes	\$1,000,000	per Tool		New, process upto 8" wafers, Cost per minutes is based 1 year of use
Wet Oxidation Furnace	228.3	per run	\$500,000	per tube	100 wafers per run	New, each run take 5 hours including ramping and cooling, Cost per minutes is based 1 year of use
Bare Silicon Wafer	13.46	per wafer				

3. The use of an ion accelerator like in Ref. 1 introduces drawbacks in terms of supply chain logistics, production line ownership costs, and ease of integration. Outsourcing the master mold production to a limited number of research facilities worldwide presents substantial risks for supply chain reliability. To have full control over the production process, one would need to own an ion accelerator, which can be prohibitively expensive. In contrast, the chemicals and tools required for silicon mold fabrication are widely available at universities, national

labs, and through external contractors, facilitating easier integration into existing production facilities. Silicon fabrication methods enable production line owners to directly convert bare silicon wafers into molds for producing superblack PDMS, reducing reliance on external facilities and providing greater control over the manufacturing process.

4. We would also like to note that the Cashew Nut Shell Liquid (No. 91) used in Ref. 1 is not listed in the U.S. Environmental Protection Agency's Toxic Substances Control Act (TSCA) chemical substance inventory. As a result, CASHEW COMPANY LTD. is unable to export this chemical from Japan to the United States (see evidence below). Therefore, the inclusion of Cashew Nut Shell Liquid could introduce additional challenges in material fabrication within the U.S. and potentially other countries.

[REDACTED]

We believe that the above information summarizes the main contributions of our work. We agree with the reviewer that the contributions of this work are not clearly articulated in the original manuscript, so we have made several improvements as described above and summarized below.

Changes made in response to comment:

1. We removed "significant improvement over state-of-art".
2. In the main text, we acknowledge that the limitation of current materials to have smaller absorbing spectrum. Also, we acknowledges that the reflectance at near normal incidence is worse than what was reported in Ref. 1.
3. We added an SEM cross-sectional image showing the steep entrance aspect ratio in the revised Figure 3g.

4. We took a bigger SEM images of the mold and increased the number of castings to over 5 times. We included the SEM images in the revised supplemental information, Figure S1.
5. We measured the repeatability of our PDMS samples by comparing several castings from the same silicon wafer (Figure S3).
6. We added a section titled “Discussion and Outlook” at the end of the manuscript in which we discuss limitations and potential for further improvement using our approach.
7. We included production considerations and cost analysis in the supporting information.

2. Retroreflection is not necessarily a drawback for most applications of supreme black materials, such as straylight. The authors should articulate why retroreflection needs to be targeted.

Response:

We agree with the reviewer that for many applications low retroreflection may not be necessary. However, for applications such as camouflaging, low retroreflection properties is important.

Changes made in response to comment:

1. We changed the text to read “Retroreflection may not be a significant drawback for some applications. However, reducing retroreflection is crucial in applications where preventing or complicating object identification with visible light is important, particularly when the object exhibits significant curvature or is viewed from glancing angles.” in the maintext.

3. Also, for most applications of supreme black materials, low reflectance over the entire UV, visible and near infrared range is required. The authors should discuss the applicability of the current method in broadband black materials.

Response:

We thank reviewer’s comment regarding the sample reflectance for the UV and IR portions of the spectrum. The intended application of this material is specifically in the visible portion of the spectrum. Based on previous work, we expect that changing the dye or pigment in the PDMS will modify the spectral response. Specifically, we expect that replacing nigrosine with carbon black will lower the reflectance in the IR. However, we also expect this to impact the reflectance in the visible portion of the spectrum as discussed in Ref. 1.

Changes made in response to comment:

1. We added the spectral limitations of the current product in the main text.

4. The durability test is insufficient. The high pressure in this manuscript is due to the small area of the tweezer tip. The impact area is too small to assess the durability. Finger touch or roller duster test should be added along with reflectance spectra before and after the durability test.

Response:

We appreciate reviewers' concerns regarding insufficient durability test. The reason that we choose tweezer tip scratching as the primary method for surface durability is the experimental reproducibility, as a round head stainless steel tweezer is nominally similar throughout the world. Finger touch reproducibility can be affected by individual dermatology conditions, as well as the cleanliness of the individual's fingers at the time of experiment. Roller duster name can be misleading depending on the country region. In Japan where Ref. 1 experiments were performed, roller duster is largely made with silicone (silicone roller duster) which is what was used in Ref. 1. However, in the U.S., roller duster is largely made with scotch adhesives. Further, we would like to mention that damaged region using a tweezer can be easily observed in black silicon as shown in Figure R13c.

Upon reviewer's request, we performed durability test using clean finger touch (Finger of Y.Y.) with a load greater than 360 g which is the maximum load of the scale. We also performed roller duster with scotch adhesives (Scotch-Brite Lint Roller) test with a load greater than 340g as shown in Figure R13. Each durability testing was done 6 times on one sample which is greater than one time (Ref. 1) with a load similar or greater than Ref. 1. The videos of these experimental demonstration (tweezer scratch, roller duster with scotch adhesives, and finger touching) are included.

Figure R13. Surface durability test using (a) finger touch (b) scotch roller duster, and (c) tweezer scratch.

We performed reflectance measurement after finger touching, roller duster with scotch adhesives, and tweezer scratches (Figure R14). We present both linear and log scale in y axis. Although Ref. 1 used log scale for their durability measurement, small variation is not as obvious in log scale. Thus, linear scale plot is included in main text.

Figure R14. Reflectance measurement of the microcavity PDMS after robustness tests.

Changes made in response to comment:

1. The presented data is included in the main text along with corresponding text.